Microbial diversity of extreme habitats in human homes

Savage Amy M. amy.savage@rutgers.edu 1
Hills Justin 2
Driscoll Katherine 3
Fergus Daniel J. 4
Grunden Amy M. 5
Dunn Robert R. 6
1 Rutgers, The State University of New Jersey , Camden , United States
2 Laboratory of Cellular and Molecular Biology, National Institute of Diabetes and Digestive and Kidney Diseases , Bethesda , MD , United States
3 Animal Management Department, The Wilds , Cumberland , OH , United States
4 Genomics and Microbiology, North Carolina Museum of Natural Sciences , Raleigh , NC , United States
5 Department of Plant and Microbial Biology, North Carolina State University , Raleigh , NC , United States
6 Department of Applied Ecology and Keck Center for Behavioral Biology, North Carolina State University , Raleigh , NC , United States
Rodriguez-Lanetty Mauricio
Electronic publication date: 2016 Sep 13
Publication date: 2016
Volume: 4
Electronic Location ID: e2376
Received 2016 Mar 16; Accepted 2016 Jul 29
Copyright: ©2016 Savage et al.
Copyright year: 2016
Copyright holder: Savage et al.
License: This is an open access article distributed under the terms of the Creative Commons Attribution License, which permits unrestricted use, distribution, reproduction and adaptation in any medium and for any purpose provided that it is properly attributed. For attribution, the original author(s), title, publication source (PeerJ) and either DOI or URL of the article must be cited.
License URL: https://creativecommons.org/licenses/by/4.0/

Keywords: Community Ecology, Extreme environments, Human Homes, Interactive effects, Microbial diversity, Temperature, pH, Chemical

Funding: A.P. Sloan Microbiology of the Built Environment Program This work was funded by A.P. Sloan Microbiology of the Built Environment Program grant awarded to RRD. The funders had no role in study design, data collection and analysis, decision to publish, or preparation of the manuscript.

==============================
High-throughput sequencing techniques have opened up the world of microbial diversity to scientists, and a flurry of studies in the most remote and extreme habitats on earth have begun to elucidate the key roles of microbes in ecosystems with extreme conditions. These same environmental extremes can also be found closer to humans, even in our homes. Here, we used high-throughput sequencing techniques to assess bacterial and archaeal diversity in the extreme environments inside human homes (e.g., dishwashers, hot water heaters, washing machine bleach reservoirs, etc.). We focused on habitats in the home with extreme temperature, pH, and chemical environmental conditions. We found a lower diversity of microbes in these extreme home environments compared to less extreme habitats in the home. However, we were nonetheless able to detect sequences from a relatively diverse array of bacteria and archaea. Habitats with extreme temperatures alone appeared to be able to support a greater diversity of microbes than habitats with extreme pH or extreme chemical environments alone. Microbial diversity was lowest when habitats had both extreme temperature and one of these other extremes. In habitats with both extreme temperatures and extreme pH, taxa with known associations with extreme conditions dominated. Our findings highlight the importance of examining interactive effects of multiple environmental extremes on microbial communities. Inasmuch as taxa from extreme environments can be both beneficial and harmful to humans, our findings also suggest future work to understand both the threats and opportunities posed by the life in these habitats.

Introduction

The innovation of culture-independent, high-throughput sequencing techniques has facilitated the discovery of high microbial diversity in many habitats once considered inhospitable to life (Rothschild & Mancinelli, 2001). The species in these environments are frequent targets for the discovery of useful enzymes (Niehaus et al., 1999; Van den Burg, 2003; Elleuche et al., 2014), and studies of microbes living in extreme environments have provided key insights into the evolution of microbial metabolism (Valentine, 2007; Hoehler & Jorgensen, 2013). Often overlooked, however, is that the attributes that define many of the most extreme habitats on Earth, such as extremes of temperature, pH, water activity, or low nutrient levels, can also be found more immediate to everyday experience. Human homes, for example, contain microhabitats as hot, acidic, basic or salty as any encountered elsewhere on Earth (Martin et al., 2015).

We know of only two extreme habitats within homes where microbial diversity has been studied to date, and in both cases culture-dependent techniques were used. In 1973, Brock and Boylen discovered a species of the genus Thermus (T. aquaticus) living in hot water heaters. Species of this genus had previously been known only from hot springs (Brock & Boylen, 1973). In addition, studies have considered the biology of tap water. Tap water is hospitable in terms of its abiotic conditions (e.g., temperature, pH, toxicity) but is very low in nutrients and so was long assumed to be relatively devoid of life; until, that is, it was studied. Tap water has now been shown to contain many species of bacteria capable of surviving in low nutrient environments (Kalmbach, Manz & Szewzyk, 1997; Szewzyk et al., 2000; Boe-Hansen et al., 2002). If life exists in hot water heaters and tap water, it seems possible and even likely that many extreme habitats in homes sustain life. That the environmental extremes imposed by these conditions in homes (cold, hot, acidic, alkaline, wet or dry) delineate which species are present seems inevitable. That they are lifeless is unlikely.

Here, we used culture-independent, high-throughput sequencing to address the following questions: (1) What is the relative diversity of microbes (specifically, Bacteria and Archaea) under extremes of temperature, pH and chemical environments of southeast US homes and how does it compare to habitats without each extreme conditions? Harrison et al. (2013) recently argued that because many extreme environments include simultaneous extremes in multiple environmental factors, interactive effects of these multiple sources of extreme conditions are likely to be important determinants of microbial diversity in extreme environments. Therefore, we additionally asked (2) how do multiple, simultaneous extreme conditions influence microbial diversity in human homes? Finally, we asked (3) which bacterial and archaeal genera from the broader home (Dunn et al., 2013) fail to persist in extreme home habitats, and which microbial genera persist only in these extreme habitats?

Methods

Sampling extreme home environments

We sampled extreme environments in six houses in the Raleigh-Durham metropolitan area (Fig. S1). In each house, we used dual-tipped sterile BBL™ CultureSwabs™ or 50 ml conical tubes to collect water from each of 10 standardized extreme locations in homes. The sites sampled in all six houses included environments that were extreme in terms of their temperature, pH and/or chemical environments (Table S1). Our assumptions concerning these sampling locations are based upon publicly available consumer resources regarding certain commercial and industrial requirements (e.g., http://www.nsf.org/consumer-resources/health-and-safety-tips/home-product-appliance-tips/sanitizing-dishwasher, http://energy.gov/energysaver/projects/savings-project-lower-water-heating-temperature). For example, our sampling of dishwashers was influenced by the NSF/ANSI 184 standard for residential dishwashers to provide a final rinse at a temperature of at least 150 °F (65.6 °C). Additionally, temperature ranges for residential water heaters are 90–150 F (32–65.6 °C), depending on the manufacturer. Bleach receptacles in clothes washing machines would also be assumed to have a pH of 12 when bleach is present. Although the pH and chemical composition of laundry detergent and dishwasher detergent can be quite variable, manufacturing standards are generally within the 7–10 pH range. While measurements, opposed to assumptions, would be very useful, taking measurements of all the potential extreme axes under various sample sites in multiple homes was not feasible. All samples were preserved at −20 °C immediately after collection.

Isolating and identifying microbes in extreme home environments

Genomic DNA was extracted from all samples using the MoBio Power Soil DNA extraction kit (MoBio, Carlsbad, CA) as described previously (Fierer et al., 2008; Lauber et al., 2009). For swabs, the tips were placed in PowerBead tubes containing solution C1 and swirled vigorously for approximately 10 s to release contents and removed. Water samples were thawed and filtered using Corning 50 ml 0.22 um cellulose acetate filters after which the filters were added to the PowerBead tubes. The extractions were subsequently performed as directed by the manufacturer, except that the final elution was performed in 50 µl of 70 °C C6 elution buffer. Because the water samples were frozen prior to filtering and extraction, the results reported for the water samples likely under-represents the true diversity of taxa in those environments.

We used methods described in Bates et al. (2011) to amplify bacterial and archaeal DNA from the samples collected from homes and six negative controls. Briefly, amplicons were produced by PCR with universal bacterial/archaeal 515F and 806R primers to which Roche 454 B pyrosequencing adapters had been added, as described in Hulcr et al. (2012). The 515F primer contained an additional 12-bp barcode sequence for individual sample identification. All the samples were amplified by triplicate PCR reactions, cleaned using the UltraClean-htp 96-well PCR Clean-up kit (MoBio), and quantified with a Quant-iT PicoGreen dsDNA Assay kit (Invitrogen). Equimolar amounts of each sample were pooled into a single sample to sequence. DNA pyrosequencing was performed at Selah Clinical Genomics Center at Innovista (University of South Carolina, USA) using a Roche Genome Sequencer 454 FLX system to facilitate comparison to previous related work that utilized this platform (Dunn et al., 2013). Though these methods here do not distinguish living from recently dead cells, with the comparative approach used here, we presume that taxa frequently identified in one habitat but rare or absent in most others are likely surviving in the habitat from which they are frequently identified. The sequences were submitted to NCBI (SRA accession number SRP071677).

The QIIME analysis package (Caporaso et al., 2010a) was used to process and analyze the barcoded microbial (bacterial and archaeal) amplicon sequences. Sequences were quality filtered to a minimum quality score of 25 with no unambiguous bases and sorted to each sample by the 12 bp barcodes. The 454 pyrosequencing produced 197,305 reads that passed the quality screening. The sequences were grouped into Operational Taxonomic Units (OTUs) that shared at least 97% sequence similarity. A representative sequence was taken for each OTU group and PyNAST (Caporaso et al., 2010b) was employed to align these representative sequences to the Greengenes database (DeSantis et al., 2006) and the taxonomic identity of each OTU was determined using the RDP Classifier (Wang et al., 2007). Phylotypes were considered to be contaminants if they were seen in at least two of the six negative control samples. There were 152 OTUs at the genus level present in more than one negative sample, representing 9% of the total OTUs at this level. After removing contaminant sequences and singletons, the number of quality-filtered reads per sample was between 6 and 5,861 (median = 2,306). Finally, we removed any OTUs represented by 20 or fewer reads to reduce the possibility of spurious results from the sequencing process. For among samples comparisons we rarefied each to a depth of 1,000 sequences. Thus, our assessments of microbial diversity are conservative.

Analysis of the relative diversity of microbes in extreme temperature, pH and chemical environments of homes and how it compares to habitats without each extreme condition

We compared microbial species accumulation among three extreme variables in homes: temperature, pH, and chemical extremes. Temperature was classified on a scale of 1–5, with 1 representing the coldest environments and 5 representing the hottest environments. We then binned 1 and 5 into an extreme temperature category and 2–4 into an intermediate temperature category. Similarly, environments were classified as acidic, basic or neutral and then binned into extreme pH (acidic or basic environments) versus neutral environments. Finally, chemical extremes were those environments characterized by the presence of detergent, bleach, metals, ammonia, or natural gas (Table S2).

We used EstimateS v.9.1.0 (Colwell, 2013) to construct individual-based species accumulations for all three extreme environments and their non-extreme counterparts. For these curves, reads were used as individuals and the curves were constructed using 1,000 iterations. To formally assess differences in accumulated species by read, we used ±95% confidence intervals for each curve. Non-overlapping 95% confidence intervals are considered formal evidence of significance (Knezevic, 2008).

Assessing how multiple, simultaneous extreme conditions influence microbial diversity in human homes

We were interested in testing the hypothesis that interactive effects of multiple, simultaneously extreme environmental conditions are important determinants of microbial diversity in extreme home environments (Harrison et al., 2013). Our study included multiple samples with more than one environmental extreme (Table S1); however, we only had sufficient replication to assess this hypothesis for 2-way interactions between extremely high temperatures and extreme pH as well as high temperature and chemical environments. Because number of reads varied significantly among different environmental extremes, we could not use a standard 2-way ANOVA. Instead, we assessed these effects using an ordination framework.

We visualized the composition of bacteria and archaea from extreme habitats in homes using non-metric multidimensional scaling ordination (NMDS) in Primer-E v.7.0.9 with PERMANOVA + 1 (Clarke & Gorley, 2015). To do this, we first constructed NMDS plots with 100 restarts and a Type I Kruskal fit scheme based on a Dissimilarity matrix of Bray-Curtis distances. To assess the relationship between temperature (extreme vs. intermediate) and the other extremes (pH: extreme vs. neutral; chemicals: extreme vs. none) for α-diversity of microbes, we conducted a permuted multivariate analysis of variance (PERMANOVA) test with temperature class and either pH or chemical class and their interaction as factors, 9,999 iterations and Type III sums of squares. When interactions were significant (Anderson, Gorley & Clark, 2008), we conducted pairwise PERMANOVA to determine which treatment combinations significantly differed from one another. Similarly, we assessed these relationships in terms of β-diversity using a permuted dispersion (PermDisp) test of a presence/absence matrix of OTU occurrences. When these tests were significant, we conducted pairwise tests of extreme vs. non-extreme chemical and pH environments in habitats with intermediate and extreme temperatures (thus 2 tests per treatment combination). Finally, we conducted SIMPER analyses for each significant treatment combination to determine the OTUs that contributed the most to pairwise between-group differences in ordination space. Because we conducted two separate analyses for each level of diversity, we accounted for the additional error associated with multiple tests, using a revised α = 0.05∕2 = 0.025 as our cut-off for statistical significance for the results of each test. This conservative α is particularly important because we did not have equal sample sizes in all groups for these analyses, which can increase the risk of Type I error (Anderson & Walsh, 2013).

Determining which microbial genera differentiate extreme home habitats from the rest of the home

We compared the occurrences of microbes in our samples to those reported in less extreme home environments (Dunn et al., 2013). Human-associated microbes were common to both datasets, and we were particularly interested in those taxa unique to our dataset, relative to the broader home (Dunn et al., 2013). Therefore, we removed human-associated OTUs from our dataset. We identified these human-associated OTUs using databases that identified human gut (Flores et al., 2014) and skin (Urban et al., 2016) microbiomes. OTUs that occurred in at least 80% of the samples in those databases were considered human-associates and excluded from our analyses of the microbial diversity of extreme habitats in human homes. We then determined the identity of microbes that were absent from the broader homes dataset, but present in extreme environments and then tabulated the extreme habitat(s) in which they were present. Likewise, we identified the non-human associated microbes that were present in the broader home environment, but absent from all extreme environments in our samples.

Figure 1 OTU accumulation curves for each extreme environment, expressed as the number of OTUs by the number of reads from sequencing.

Each curve was constructed using 1,000 iterations, and the dotted lines represent 95% confidence intervals. Non-overlapping confidence intervals indicate that the accumulation curves are significantly different. Thus, habitats with extreme temperatures had significantly more accumulated species than habitats with either extreme pH or extreme chemical environments. However, the accumulated species in habitats pH and chemical extremes did not differ significantly.

Figure 2 Comparison of rarefaction curves between extreme and non-extreme habitats.

(A) Extreme vs. intermediate temperatures, (B) extreme vs. neutral pH environments, and (C) extreme chemicals present vs. absent. Rarefaction curves are expressed as number of OTU by number of reads from sequencing. Each curve was constructed using 1,000 iterations, and the dotted lines represent 95% confidence intervals. Significance tests were as described for Fig. 1.

Results and Discussion

What is the relative diversity of microbes in extreme temperature, pH and chemical environments of southeast US homes and how does it compare to habitats without each extreme condition?

The cumulative diversity (OTU richness) in habitats with extreme temperatures was more than twice as high as in habitats with extreme pH (maximum of 73 vs. 33, Fig. 1) and almost three times as high as habitats with extreme chemical environments (27.6; Fig. 1). Habitats with extreme temperatures also had higher OTU richness than habitats with intermediate temperatures (Fig. 2A). Conversely, previous research indicates that the diversity in habitats with either extremely high or extremely low temperatures is generally low, and dominated by a small number of abundant bacterial species (Lewin et al., 2013). For example, Sharp et al. (2014) recently found that OTU richness in hydrothermal vents peaked at intermediate temperatures (24 °C), with reduced OTU richness in extremely hot or cold environments (Sharp et al., 2014). We did not detect significant differences in the rarefied species richness of bacterial and archaeal microbes in extreme vs. neutral pH conditions; however, the marginally non-significant trend suggests that extreme pH environments also had higher microbial diversity than neutral habitats (Fig. 2B). Recent studies have demonstrated that pH is a key predictor of microbial diversity in both extreme environments, such as acid mine drainage sites (Kuang et al., 2013), and less extreme environments, such as tropical soils (Tripathi et al., 2012). In both cases, habitats with neutral pH had higher microbial diversity than those with a pH higher or lower than neutral. Thus, we again found different patterns in extreme home environments compared to other studies comparing extreme and non-extreme habitats. One possible explanation for the difference between our findings and these recent studies is that human-associated microbes are present in home environments with intermediate temperatures. Perhaps these species are able to dominate habitats with intermediate, but not extreme, conditions. Alternatively, the lower diversity in habitats with intermediate temperatures and neutral pH in our study could be due to the occurrence of extreme conditions along different axes (e.g., intermediate temperature, but extreme pH or chemical habitats). We examine potential interactive effects of these polyextreme habitats in the next section.

In contrast, habitats with extreme chemicals had significantly lower accumulated OTU richness than did habitats without these extreme conditions (Fig. 2C). Extreme chemical environments are poorly studied and understood (Rothschild & Mancinelli, 2001). Our data suggest that they could act as strong filters in extreme environments.

How do multiple, simultaneous extreme conditions influence microbial diversity in extreme home environments?

Many of the habitats in this study were characterized by more than one extreme environmental condition. Therefore, we also examined the potential for interactive effects of multiple, simultaneous extreme conditions on microbial diversity. Due to limited replication across all environmental extremes, we were only able to examine extreme pH and chemical habitats with and without extreme temperatures. We used an ordination framework to examine these interactive effects (see ‘Methods’).

We found significant interactions between extreme temperature and both extreme pH (PERMANOVA: Pseudo-F1, 82 = 2.53, P = 0.0001; Fig. 3A) and extreme chemical (PERMANOVA: Pseudo-F1, 82 = 3.16, P = 0.0001; Fig. 3C) environments for OTU composition. When temperatures were intermediate, there were no significant differences in microbial composition in extreme vs. neutral pH habitats (pairwise PERMANOVA: t1, 38 = 1.02, P = 0.40). However, when temperatures were extreme, there was a very large difference between the composition of microbes in extreme pH habitats, compared to neutral habitats (pairwise PERMANOVA: t1, 38 = 1.70, P = 0.0001; Fig. 3A). The five genera that contributed the most to differences between these two habitat types (from SIMPER analysis) were Parascardovia, Micrococcus, Rothia, Brachybacterium, and an unknown genus from Sphingomonadaceae. Most of these genera are associated with humans (Oshima et al., 2015; Gueimonde et al., 2012; Kloos & Musselwhite, 1975; Kocur, Koos & Schleifer, 2006; Vaccher et al., 2007; Uchibori et al., 2012). Sphingomonadaceae are widespread in aquatic habitats, including drinking water (Vaz-Moreira, Nunes & Manaia , 2011), but also other aquatic environments (e.g., tree holes-Xu et al., 2008). Brachybacterium is usually associated with marine environments (Ward & Bora, 2006), including Antarctic sea ice (Junge et al., 1998). However, it was recently detected in an urban shopping center (Tringe et al., 2008). All of these genera were more common in habitats with extreme temperatures and neutral pH than they were in habitats with both extreme temperatures and extreme pH. While different houses had significantly different microbial composition (3-way PERMANOVA, P = 0.0001), there were no significant 2-or 3-way interactions with house (Table S3).

Figure 3 NMDS ordinations OTU occurrence by (A–B) temperature & pH and (C–D) temperature & chemical environments in the home.

Large symbols represent centroids ±1 SE (A, C), and small symbols represent each sample (B, D). The interaction between temperature and pH was significant (PERMANOVA: (pseudo)-F1, 85 = 2.53, P(perm) = 0.0001), as was the interaction between extreme temperature and chemical conditions (PERMANOVA: (pseudo)-F1, 85 = 3.16, P(perm) = 0.0001). The ordination was constructed with Bray-Curtis distances and 100 restarts; 2-D stress was 0.21. PERMANOVA analyses were conducted using type III sums of squares and 9,999 iterations.

The interaction between temperature and chemical extremes was similar. Microbial composition was indistinguishable between the habitats that only had one extreme condition-regardless of whether it was temperature or chemicals that were extreme. There were also no significant differences between habitats with neither extreme temperatures nor extreme chemical conditions and habitats that had a single extreme condition. However, habitats with both extreme temperatures and extreme chemicals had significantly different microbial composition compared to all other groups (pairwise PERMANOVA; t1, 38 = 1.75, P = 0.0001; Fig. 3C). The five genera that contributed the most to compositional difference between these two habitats (from SIMPER analysis) were Methylobacterium, an unknown genus of Moraxellaceae, Sejonia, an unknown genus of Sphingomonadaceae, and Flavobacterium. With the exception of the unknown genus of Moraxellaceae, which was more common in extreme chemical and temperature environments, all of these genera were more common in the habitats without temperature and chemical extremes. Moraxallaceae have been found in other extreme environments, including deep sea sediments (Maruyama et al., 1997). Although it was more common in our less extreme environments, Sejonia is better known from Antarctic ice (Yi, Yoon & Chun, 2005). Sphingomonadaceae as described above are common to aquatic habitats. Methylobacterium is a widespread habitat generalist that is facultatively methyltrophic (Green, 2006). Finally, Flavobacterium is common in freshwater and marine ecosystems but tends to flourish in cold environments with high salinity (Bernardet & Bowman, 2006).

Figure 4 Average distances between samples and centroids (β-diversity) across home environments that differ with respect to extreme temperatures and (A) extreme pH conditions & (B) extreme chemical conditions.

Data were assessed using PermDisp; dispersion was significantly different across extreme temperatures and extreme pH conditions (F3, 82 = 4.08, P = 0.024) and across extreme temperatures and extreme chemical conditions (F3, 82 = 6.99, P = 0.0017). Post-hoc pairwise tests: * P < 0.025, ** P < 0.01, *** P < 0.001.

There were also significant differences in the β-diversity in home environments with more than one extreme condition. When temperatures were intermediate, there were no significant differences between neutral and extreme pH environments (Fig. 4A; PermDisp: P = 0.3864). However when temperatures were also extreme, habitats with extreme pH conditions had significantly higher β-diversity than those with neutral pH conditions (Fig. 4A; PermDisp: P = 0.0014). Similarly, at intermediate temperatures, there was a non-significant trend (Fig. 4B; PermDisp: P = 0.03, Bonferroni-corrected α = 0.025) in which habitats without extreme chemicals present had higher β-diversity than those with extreme chemicals present. However, when temperatures were also extreme, habitats with extreme chemicals present had higher β-diversity than those without extreme chemicals (Fig. 4B; PermDisp: P = 0.0006). This increase in β-diversity in extreme pH and chemical environments when temperatures were also extreme suggests that polyextreme conditions may support a higher diversity of extremophiles and/or reduced occurrences of numerically dominant genera compared to environments with a single extreme condition, at least among habitats (in contrast to within habitats). The 5 genera that contributed the most to differences in β-diversity between neutral and extreme pH conditions when temperatures were also extreme were: Veillonella, Kocuria, Peptoniphilus, Parascardovia, and Anaerococcus. Interestingly, these were also the top 5 genera contributing to differences between habitats with and without extreme chemicals that also had extreme temperatures. All of these genera were less common in habitats with 2 extremes than they were in habitats with only extreme temperatures. They are also genera that include human-associated species (Bhatti & Frank, 2000; Fadda, Vignolo & Oliver, 2001; Song, Liu & Finegold, 2007; Gueimonde et al., 2012).

Table 1 Summary of occurrences of microbes that were present in samples from extreme home environments, but absent from the broader home samples.

Numbers indicate the number of reads of each genus by extreme environment. The first group includes genera that were only present in one extreme environment, the second group includes genera that were present in two extreme environments, and the last group includes genera that were present in all three extreme home environments.

Genus	Extreme temperatures	Extreme pH	Extreme Chemical	
Brochothrix	265	0	0	
Buchnera	22	0	0	
Polynucleobacter	33	0	0	
Ralstonia	21	0	0	
Thermicanus	34	0	0	
Helcococcus	0	0	22	
Solibacter	86	0	30	
Brevundimonas	184	189	0	
Azobacteroides	0	33	33	
Elizabethkingia	0	25	24	
Xiphinematobacter	0	19	21	
Azospira	139	33	44	
Brachybacterium	101	52	69	
Enhydrobacter	452	387	408	
Gluconobacter	23	21	22	
Oligella	40	74	77	
Parascardovia	141	46	107	
Photobacterium	71	65	93	
Propionibacterium	73	31	40	
Salinibacterium	108	334	355	

Which microbial genera differentiate extreme home habitats from the rest of the home?

After removing all human-associated microbes (see ‘Methods’), there were a total of 241 unique genera in the broader homes dataset (Dunn et al., 2013). Our extreme samples contained 135 of the remaining broader homes genera, but ∼44% of the genera found in the broader homes were absent from our extreme home samples (Table S4), the absence of which might simply be due to the larger number of samples in Dunn et al. (2013). More interestingly, we found 20 genera present among our samples that were absent from the broader homes dataset. Nine of these genera were found in all three categories of extreme environments (Table 1); one genus (Solibacter) was absent from habitats with extreme pH, but occurred in both extreme chemical and temperature environments. Solibacter is a common and abundant soil microbe, especially in tropical regions (Guan et al., 2013; Wang et al., 2015). There was also one genus (Brevundimonas) that was absent from extreme chemical environments, but present in both extreme temperature and extreme pH environments; Brevundimonas is one of the only genera thought to be able to survive the low temperatures and ionizing radiation on Mars (Dartnell et al., 2010). There were three genera (Azobacteroides, Elizabethkingia, and Xiphinematobacter) that occurred in both extreme pH and chemical environments that were absent in extreme temperature environments. Both Azobacteroides and Xiphinematobacter are gut symbionts of invertebrates; Azobacteroides is commonly found inside the protozoan symbionts of termites (Noda et al., 2007), and Xiphinematobacer is an endosymbiont of nematodes (Vandekerckhove et al., 2000). In invertebrate guts these microbes likely experience extreme chemical and pH environments frequently, while being relatively protected from temperature stress. Elizabethkingia is a cosmopolitan genus, with species that are endosymbionts of mosquitoes (Kämpfer et al., 2011), and others that are pathogens of both humans (Ceyhan & Celik, 2011) and frogs (Xie et al., 2009). There was one genus that was only found in extreme chemical environments (Helcococcus). Interestingly, members of the genus Helcococcus possess the ability to degrade detergents. In fact, the detergent Tween-80 can be added to media to enrich Helcococcus (Collins et al., 1993; Chagla et al., 1998). Finally, we found 5 genera (Brochothrix, Buchnera, Polynucleobacter, Ralstonia, and Thermicanus) unique to extreme temperature environments. Brochothrix is a common spoilage bacterium in meat (Rattanasomboom et al., 1999). Buchnera is a widespread aphid endosymbiont (Shigenobu et al., 2000). Recently, a survey of homes in Raleigh, NC demonstrated that aphids could be quite common in human homes (Bertone et al., 2016), which could explain how this genus arrived in the homes in our study (via aphids in the home). The genus Polynucleobacter includes both free-living species and species that are endosymbionts of nematodes (Vannini et al., 2007). Ralstonia metallidurans is a bacterium specifically adapted to toxic metal environments (Mergeay et al., 2003). Other species of Ralstonia have been shown to be effectively controlled using high temperature treatments in commercial crops (Kongkiattikajorn & Thepa, 2007). In our study, Ralstonia were collected in both high and low temperature environments. Finally, Thermicanus is, as its name suggests, a thermophilic bacterial genus (Wrighton et al., 2008).

Conclusions

This study has provided a glimpse into the microbial diversity that lives in habitats of human homes similar in their extreme temperature, pH and chemical conditions to some of the most extreme habitats on Earth. We discovered that these conditions have lower diversity than the surrounding home environment; yet tens of bacterial lineages can be found in these extreme habitats of the human home, including many taxa with known associations with extreme conditions. Habitats with extreme temperatures alone appear to be able to support a greater diversity of microbes than habitats with extreme pH or extreme chemical environments alone. Microbial diversity is significantly lowest when habitats have both extreme temperature and one of these other extremes. Interestingly, environments in homes often alternate between periods of extreme and non-extreme conditions. For example, dishwashers are only likely to have extremely high temperatures while cleaning and drying dishes. This variability could lead to temporal shifts in microbial composition, similar to those found for human vaginal microbes (Gajer et al., 2012). This variability may also explain the presence of human-associated generalist species in our samples. Future work, with samples taken before and after appliances (like many of those used in our study) are operated, could elucidate the importance of episodic extreme conditions for microbial communities in homes. Additionally, a key next step is understanding which of the relatively few species that are found in these poly-extreme environments in the home are metabolically active there and both whether these polyextreme taxa pose health threats (as was recently suggested by Gümral et al., 2016) and/or might be useful industrially.

Supplemental Information

Data S1 Raw data from sequence analyses

Click here for additional data file.

Supplemental Information 1 Supplementary Tables 1–3 & Supplementary Figures 1–3

Click here for additional data file.

Supplemental Information 2 List of non-human associated microbes in extreme and non-extreme (Dunn et al., 2013) home habitats

Click here for additional data file.

We thank the homeowners for allowing us to sample habitats in their homes. Thanks also to Dr. Holly Menninger for her support and guidance in addressing logistical challenges presented by the current work. The Genomics Laboratory at the North Carolina Museum of Natural Sciences provided critical logistical support for this project.

Additional Information and Declarations

Competing Interests

Author Contributions

DNA Deposition

Data Availability

The authors do not have any competing financial or non-financial competing interests to report.

Amy M. Savage analyzed the data, contributed reagents/materials/analysis tools, wrote the paper, prepared figures and/or tables.

Justin Hills and Katherine Driscoll conceived and designed the experiments, performed the experiments, reviewed drafts of the paper.

Daniel J. Fergus and Amy M. Grunden contributed reagents/materials/analysis tools, reviewed drafts of the paper.

Robert R. Dunn conceived and designed the experiments, contributed reagents/materials/analysis tools, reviewed drafts of the paper.

The following information was supplied regarding the deposition of DNA sequences:

NCBI SRP071677.

The following information was supplied regarding data availability:

The raw data has been supplied as a Supplemental Dataset.

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
