# Peer review of "Microbial diversity of extreme habitats in human homes"

_PeerJ, doi:10.7717/peerj.2376_

## Round 0.1 · original submission · Major Revisions

· Academic Editor

Major Revisions

There are important suggestions regarding the presentation and analysis of the graphical data, which requires revision – these include the nMDS analyses, and rarefaction plots.

I would encourage the authors to address all the suggestions by the reviewers and pay careful attention to some of the questioning of the hypotheses and address the lack of some methodological details raised by the reviewers.

Reviewer 1 ·

Basic reporting

The paper reports the bacteria identified in particular places in the home that are marked by potentially unfavorable conditions - temperature, pH, chemical composition. While the study questions and design are valid, there are issues with presentation.
-- Microbial diversity includes more than bacteria, but this paper only sampled bacteria. References to microbial should be changed to bacteria for most of the paper, while microbial used when the focus is broader than bacteria.
-- The actual environmental conditions of the sampling locations were not measured but (logically) presumed. Supp Table 1 categorizes locations, but these are based on assumed conditions, and the Introduction talks about water heaters and tap water (temperature is the extreme) but not about pH and chemistry. Since the paper is centered on the idea that bacterial life probably thrives in extreme indoor environments as it has in extreme outdoor environments, the connection between extreme environments in homes and outdoors could be much more rigorous and based on measurements rather than assumed conditions. It also seems that the extreme environments in homes are transiently extreme (for example, high temp water only when the washing machine is running, and on the warm setting), while in the outdoors, habitats are more consistently extreme (are hot springs ever moderate temperature?). This variability in extreme conditions should be included in the framework for how it would affect bacterial growth.
--A focus of the paper is about the richness of bacteria found in the sampled locations. However, the comparison between extreme habitats and non-extreme habitats is never made, although those data seem to be available. If choosing not to explore those comparisons, the reasons should be stated.

Experimental design

--What are the variance explained (R2) with the different PerMANOVA analyses (starting Line 127)? While significant, they may only explain a small portion of the variation in bacterial communities, and this is an important aspect to consider.
-- Line 256-258: Its great that negative controls were used, although it would be state how many OTUs were considered to be contaminants, just to give the reader some idea. Were the samples rarefied or standardized for read number in some way, for when comparing across samples?

Validity of the findings

--On lines 108-113, there are two hypotheses offered as to why richness in extreme home conditions may be different than outdoor conditions. The first, that the lower richness in intermediate temperatures is because you removed human-associated taxa, could be tested with the data that you have: prior to removing human-associated bacteria, compare the rarefaction plots. The second hypothesis isn't clear to me, and is seemingly contradictory - the same habitat has an intermediate temp and a neutral pH, and a intermediate temp and extreme pH. Also, aren't these habitats across multiple axes specifically tested in the next section?
-- Line 135: Its confusing when human-associated taxa were removed, and then its mentioned that most of the genera separating temperature (or pH, not sure) were associated with humans.
-- Comparing extreme home habitats from the rest of the home is an important part of the paper. Because you are comparing across data sets, its important to rule out differences due to sequence runs. Its interesting the you found 20 genera in the extreme habitats that weren't present in the larger home, although I'm left wondering how abundant (within samples) or frequent (across samples) they were. For example, a taxon might have been found in one high temperature appliance in one home but not found in the larger dataset, and it would get a check mark in Table 1.
-- Line 193-194: Why would animal endosymbionts be found in high temperature environments in the home?

Additional comments

Some minor editorial comments:
Line 55: Odd wording - species are "key insights"
Line 73/74: Repetitive from last sentence of earlier paragraph.
Line 80: "Additionally" confused me
Line 108: unclear what "two studies" are referenced here
Line 133: which two habitat types - temperature gradient or pH?
Supp Table 1: would be helpful to specify which types were swabs and which were conical tubes. Were the filters swabbed?
Line 306: Does removal of OTUs less than 20 times mean that it had to have a total abundance across all samples (even if it was just 1 sample) of 20, or it had be found in 20 samples?
Line 310: extra period.

·

Basic reporting

The article meets the basic reporting standards of the journal

Experimental design

Savage et al. aim to address an important question in the microbial ecology of the built environment by characterizing the microbial diversity of “extreme” and understudied environments within residential homes. They have collected an interesting dataset that includes 22 different sample types across 6 individual homes that span a wide range of temperatures, pH, and chemical compositions. Both their questions and their dataset are interesting, which is why I was disappointed by the relatively superficial analyses of the paper.

Validity of the findings

The authors confine their results almost exclusively to rarefaction curve-based analyses of alpha diversity and to a discussion of the presence/absence of various genera in different environmental conditions. None of these analyses are performed incorrectly, and their conclusions are valid, but many more interesting questions could be addressed with a dataset of this size.

Additional comments

Sample types are coarse-grained into broad environmental types, and the fact that this dataset comprises 22 different sample types in 6 different homes is lost entirely. In fact, the sample types are not mentioned at all in the main text and are confined to the supplement. There is no discussion of, for example, beta diversity between samples, and the relative importance of home vs. sample type in shaping extreme communities.

It seems the authors are wary of rarifying their dataset to an even level because so many sequences would be dropped in the process. This is an unfortunate problem in every metagenomic study, but given that the authors report a median depth of 2300 reads, it seems that they would retain most of their samples rarifying to 1000 or so reads. This would allow them to answer many questions about the relative impacts of different factors on diversity that are only answerable with beta-diversity calculations.

I am also unclear on why the authors have chosen to remove human-associated OTUs from the study. Almost all studies of the built environment have shown human skin to be the dominant source of residential microbiota. This may be less true in these extreme environments, but it would be interesting to see how the proportion of diversity that seems to originate from occupants varies across different sample types. Since the authors already know how many sequences were removed from each sample, they already have this information, and I suggest they report it.

Other comments: The y-axis of figure 1 is incorrect

Reviewer 3 ·

Basic reporting

Line 91: "rarefied OTU richness" - what metric is used here? This should be clarified. The authors also refer to this as species richness (line 100). The comparisons that the authors are describing are not adequately reported. For example, "(73 vs. 33)" - what is the sample size for each group? What test was used to compare these? Is the difference statistically significant (sounds likely, but we don't know based on what the authors have presented here).

Figure 1 legend: "number of OTU" should be "OTUs". The y-axis is labeled incorrectly in this plot, and there are some formatting issues with this legend. The number of samples in each group needs to be reported, and again, the metric that is being used for computing richness.

Overall, the figures are inadequate. Why are the authors using rarefaction plots to compare diversity at a single rarefaction level? Nothing is being said about depth of coverage. Distribution plots (e.g., histograms) and associated statistics would be far more relevant for these comparisons. Similarly, why are all of the extreme environment samples being grouped in NMDS? It's far more useful to see the spread of all of the samples directly. The NMDS confidence interval plots should be replaced with standard NMDS plots that show all of the samples, and statistics that describe groupings of samples (e.g., PERMANOVA, PERMDISP, ...) should be presented with the figure. They should also describe what metric is being used here ("NMDS ordinations OTU occurrence" is not a useful description). The word limits are lenient, so this information doesn't need to be buried in the methods. Finally, any time a distribution is being summarized, as is being done in all figures, the sample size needs to be reported.

"PerMANOVA" - I don't think this is the correct capitalization. It also differs throughout the text. PERMANOVA also doesn't detect interactions (line 126), but rather grouping.

Lines 133, 150: What test was used for differential abundance detection? The authors should present the median abundance and test statistics for each taxon.

Figure 4 legend typo "Extrme".

Experimental design

Are the "special locations" (most of which have n=1) included in the distribution summaries presented in the figures?

What are the "broader home" samples? Those are not defined in the supplementary table. Why aren't these compared in Figure 1 (which would be important for the statement on line 36) and 3a?

Similarly, more detail is needed on the dropping of human samples. How were human-associated OTUs identified?

Validity of the findings

Without the sample sizes and the statistics I mention, it's not possible to determine if these findings are valid.

Any mention of comparison to non-extreme environments needs clarification on what samples are being compared.

Additional comments

Why the mention of pathogens on line 46? It doesn't seem like there is an evidence from this work that the authors have identified pathogens in the home extreme environments. I think this should be dropped. As it stands, it's effectively saying there might be pathogens in home extreme environments, but we didn't find any.

---

## Round 0.2 · Minor Revisions

· Academic Editor

Minor Revisions

The new version of the manuscript has been considerably improved, nevertheless there are still some minor questions regarding the methods and the statistical analyses used. I recommend the authors address them.

Reviewer 1 ·

Basic reporting

line 36: The pre-edited wording was more appropriate. Finding a sequence in a location does not mean that the organism is alive or using that area as a habitat. Although this idea is addressed in lines 162-165, there are real issues with using frequency as a surrogate for viability, and how the exact approach of using frequency in the analysis is actually implemented in unclear in the methods section.
Figure 3 - its helpful to see the raw values as well as the centroids, but it only makes me wonder more about the variance explained (it looks small).

Experimental design

No Comments.

Validity of the findings

--Lines 171: rarefaction was done prior to OTU clustering? Typically, rarefaction (or some approach to standardizing depth across samples) is done on the "OTU table".
--I'm generally confused by analysis choices described in 212-225. Typically, PERMANOVA is used to characterize whether the composition across sample types is different - what I think of as beta diversity. How was it used to compare alpha diversity? Then, PermDist is typically done to see if sample groups have different dispersions, and if they do, this helps interpret the results of the PERMANOVA, since different dispersions can lead to spurious significant relationships in PERMANOVA (http://www.primer-e.com/permanova.htm). Along these lines, percent variance explained can be explained using PERMANOVA. It's quite common for PERMANOVA to be significant but the variance explained to be tiny (<0.05%). I found this on ResearchGate:
Here is correspondence from Marti Anderson,"You have to convert them into percentages yourself by summing them and then taking the proportion out of the total. This approach is fine for balanced designs, but be aware that the partitioning is not unique for unbalanced designs (the order of terms will matter), so you would have to think about this carefully first (and choose an appropriate order and type of SS, etc.) before reporting percentages."

Additional comments

The manuscript is improved but some questions on methods and presentation remain.

·

Basic reporting

The paper meets the basic reporting standards of the journal.

Experimental design

Although the authors are trying to answer an important question in built environment microbiology, and they present some compelling findings, the experimental design of this study limits the scope and impact of those findings. The fact that environmental extremity is inferred rather than measured, and that sampling locations are not standardized across homes, limits both the precision of the study and the ability to quantify inter-home variation (which is the focus of numerous prior studies and is almost always found to be the single most important source of variation). Still, the authors have collected an interesting dataset, their research question is clearly defined, and it focuses on a major knowledge gap in built environment microbiology.

I have a few specific questions about methods that should be addressed. I’m confused by the OTU clustering process discussed in the paragraph beginning on line 134. The authors mention rarifying the data to an even depth of 1,000 reads/sample, but they appear to have done so before filtering out contaminants clustering the reads into OTUs? They also note that the number of quality-filtered reads per sample is between 6 and 5861, which suggests that they ran clustering after quality filtering.

Line 144: “There were 152 OTUs at the L6 level” is confusing, as L6 is genus-level in QIIME. Does this mean that there were 152 genera (or unclassifiable to genus level nodes) in the dataset?

Validity of the findings

The data are analyzed appropriately and methods are clear except in the case of OTU clustering (see above).

Additional comments

The sentence beginning on line 90 and the sentence beginning on line 92 are redundant.

"Cumulated OTUs" seems like a strange choice for the Y-axis in figure 2. "Number of OTUs" would be a better choice (and is standard for most rarefaction curves I've seen in the literature).

---

## Round 0.3 · accepted · Accept

· Academic Editor

Accept

The authors have addressed satisfactorily all the comments and suggestions raised by the reviewers.